# What Matters for In-Context Learning: A Balancing Act of Look-up and In-Weight Learning

## Abstract

Large Language Models (LLMs) have demonstrated impressive performance in various tasks, including In-Context Learning (ICL), where the model performs new tasks by conditioning solely on the examples provided in the context, without updating the model's weights. While prior research has explored the roles of pretraining data and model architecture, the key mechanism behind ICL remains unclear. In this work, we systematically uncover properties present in LLMs that support the emergence of ICL. To disambiguate these factors, we conduct a study with a controlled dataset and data sequences using a deep autoregressive model. We show that conceptual repetitions in the data sequences are crucial for ICL, more so than previously indicated training data properties like burstiness or long-tail distribution. Conceptual repetitions could refer to $n$-gram repetitions in textual data or exact image copies in image sequence data. Such repetitions also offer other previously overlooked benefits such as reduced transiency in ICL performance. Furthermore, we show that the emergence of ICL depends on balancing the in-weight learning objective with the in-context solving ability during training.

## 1 Introduction

In-context learning (ICL) is a remarkable feature of Large Language Models (LLMs) (Radford et al., 2019; Brown et al., 2020) since it enables the model to adapt and solve tasks never seen during training, conditioned solely on the context provided during inference (Brown et al., 2020) without demanding any retraining or task-specific fine-tuning. ICL contrasts with standard *in-weight* learning (IWL), where the knowledge needed for inference tasks is embedded within the model weights during training. Models showing ICL are trained autoregressively to predict the next token as the in-weight learning objective. However, since no explicit training objective is tailored for ICL, it is challenging to identify the underlying factors for its emergence.

Previous research (Han et al., 2023; Chan et al., 2022) has attributed the emergence of ICL to pretraining data properties such as long-tail token distribution and high burstiness in data sequences. Here, burstiness refers to clustered occurrences of data points within a sequence. Following up on the idea of reoccurring concepts in data sequences, Chen et al. (2024) analyzed the impact of parallel structures in the pretraining data of LLMs, showing that the pairs of phrases following similar templates in the pretraining corpora lead to ICL. This paper shows that while these properties improve ICL performance, they are not the predominant factors for ICL emergence.

We distinguish key driving factors of ICL operating at the data sequence level and the in-weight learning objective level. At the data sequence level, we demonstrate that it is important for the training data to provide opportunities to solve training tasks in context, which we refer to as an in-context look-up mechanism. Specifically, in this work, we show that the look-up mechanism can be strongly driven by conceptual repetitions in data sequences, which are also commonly present in the training sequences of LLMs. In Figure 1, we illustrate that the pretraining text corpora used to train LLMs naturally contain a high frequency of repetitive n-grams in the context, indicating high concept-level repetitions. It shows that an input sequence of the context window of 2048 tokens contains more than ten 10-gram repetitions across different corpora on average. In this work, we conduct a controlled study to showcase the contribution of similar repetitions for ICL. At the in-

weight learning objective level, we hypothesize that the high complexity of the in-weight learning objective is crucial for consistent ICL performance throughout the training process. In contrast, a simple IWL objective can result in subdued or transient ICL performance. This aspect is often overlooked because the training objective in LLMs is naturally quite complex, which demonstrate strong and stable ICL performance. To support this argument, we study the impact of different IWL objectives on ICL performance.

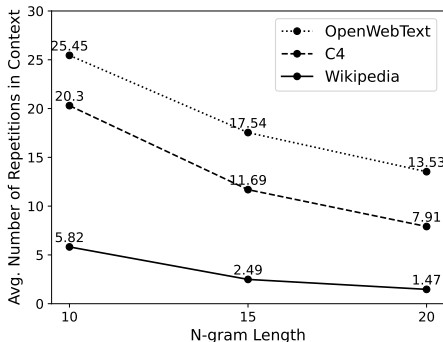

Elymus, attended Pirithous' wedding, fought in the battle against the Lapiths and was killed by... Eurytion, acted in an insulting manner towards Hippolyte when she was being joined in marriage to Azan or in the house of Pirithous... Hodites, fought against the Lapiths at Pirithous' wedding. Killed by Mopsus. Hyles, attended Pirithous' wedding, fought in the battle against the Lapiths and was killed by... Imbreus, fought against the Lapiths at Pirithous' wedding and was killed by Dryas... Isoples, killed by Heracles when he tried to steal the wine of Pholus... Lycabas, attended Pirithous' wedding, fought against the Lapiths and fled. Lycidas, fought against the Lapiths at Pirithous' wedding and was killed by Dryas. Lycus, fought against the Lapiths at Pirithous' wedding was killed by Pirithous. Medon, attended Pirithous' wedding, fought against the Lapiths and fled. Melanchaetes, killed by Heracles when he tried to steal the wine of Pholus. Melaneus, attended Pirithous' wedding, fought against the Lapiths and fled. Mermerus, wounded by the Lapiths at Pirithous'...

Figure 1: Left: Repetitions of n-grams in Wikipedia(Foundation), OpenWebText(Gokaslan et al., 2019) and C4(Raffel et al., 2020) pretraining corpora, performed over 50 million tokens using a BPE tokenizer with a context length of 2048. We report the average number of repetitions within the 2048-token window for different n-gram lengths. The variety in the corpora's format (e.g. web, news, social media, wiki) leads to substantial differences in repetition rates. Right: Truncated example of a 2048-token sample from Wikipedia's pretraining corpora, highlightingexact n-gram repetitions. Different colors present different n-gram lengths (green: 10-grams, blue: 15-grams, orange: 20-grams), demonstrating both patterns in the pretraining data.

We present a controlled study to show the contribution of each responsible factor. We train an autoregressive GPT-2 model for the image classification task where the data sequence contains input-output pairs. This data format allows us to study different data properties and training objectives. We test the impact of high and low burstiness in data, repetitions, and skewed distribution on ICL performance. We further examine the influence of different in-weight learning objectives on ICL performance by modifying the number of classes, using skewed distribution, introducing noise in the supervision signal, and shifting the in-weight learning objective to a much more complex instance discrimination task.

Our analysis shows that even a single repetition in the context sequence enables the ICL emergence without other properties like high burstiness or skewed label distribution. In our setup, we obtain strong ICL ability without significantly impacting IWL performance. Similar behavior is reported for LLMs (Brown et al., 2020; Chowdhery et al., 2024) and Vision-Language Models (Alayrac et al., 2022; Huang et al., 2023), where both ICL and IWL performances are reported even at the end of the long training period. Prior controlled studies observe a transient nature of ICL, showing diminished ICL performance as IWL training progresses (Chan et al., 2022; Singh et al., 2023) while our setup shows a much more stable ICL performance. Therefore, being a more representative model for studying ICL in large models. In our setup, ICL transiency is eliminated by using repetitions in the data sequences and using a complex in-weight learning objective.

To summarize, we contend that the strong look-up mechanism created by repetitions in the training data sequences along with a complex in-weight learning task, enable non-transient and stable ICL performance. We speculate the same mechanisms enable ICL in LLMs, where the ICL performance is stable and non-transient even after the long training phase.

## 2 RELATED WORK

**ICL research directions.** Plenty of research has been dedicated to understanding and optimizing the model response to obtain the best ICL performance. Prior works have analyzed the important

concepts that enable ICL in LLMs such as pretraining data (Liu et al. (2022); Rubin et al. (2022); Levine et al. (2022)), in-context prompt design (Voronov et al., 2024), impact of the in-context samples and length (Agarwal et al., 2024), and more. Studies (Liu et al., 2022; Rubin et al., 2022) have also focused on designing demonstrations, also known as prompt engineering, to get better responses from pretrained language models. Other works (Akyürek et al., 2023; Li et al., 2023; Wu et al., 2024) tested the ability of ICL to solve novel tasks and its robustness (Raventós et al., 2024; Min et al., 2022) by evaluating on out-of-distribution tasks. Furthermore, (Xing et al., 2024; Chan et al., 2022) aimed to understand the underlying components driving ICL, such as model architecture, training data distribution, and optimization objectives.

Simultaneously, ICL has been studied in the context of specific tasks. Many works have attempted to explain the emergence of ICL by considering it as a regression task. (Akyürek et al., 2023; Dai et al., 2023; Von Oswald et al., 2023) showed that self-attention architectures with linear attention implement a gradient descent with in-context examples, while Dai et al. (2023) theoretically identified that the transformer attention mechanism has a dual form of SGD, where in-context learners implicitly perform fine-tuning. Other works showed that transformers can implement simple functions in-context like least squares, ridge regression, and gradient descent in two-layer neural networks (Bai et al., 2023; Garg et al., 2022). Alternatively, ICL for classification tasks relies on an in-context solving ability to match and retrieve the label from the context. This demonstrates a clear difference in the underlying mechanism between these two tasks. Our work focuses on understanding the relationship between different components for the classification task setup. Given the different underlying ICL mechanisms, our findings may not directly transfer to regression tasks.

**Mechanistic perspective on ICL.**   Prior work has studied ICL from different angles through simplified experiments and model probing. Many works have analyzed the transition phases during the training process, discovering the sequence of operations and circuitry leading to ICL. Olsson et al. (2022) provided the initial evidence that induction heads may be pivotal for in-context learning in transformer-based models. Singh et al. (2024) conducted another mechanistic study using a causal approach to understand the abrupt emergence of induction heads and identified three interacting sub-circuits leading to their formation. Reddy (2024) demonstrated with a simple two-parameter model that ICL is driven by the formation of an induction head, which emerges due to nested non-linearities in a multi-layer attention network. Building on previous work in mechanistic interpretability, we analyze the emergence of induction heads alongside ICL in our setup using a small GPT-2 model with three layers and one head. Singh et al. (2023) found that ICL becomes transient as training progresses and discussed different ways to reduce transiency. In this work, we also observe similar transiency and analyze how induction heads appear and disappear as ICL becomes transient.

**Training data properties for ICL.**   Numerous works indicate that the training data distribution both in the corpus and sequences plays a role in the emergence of ICL. Shin et al. (2022) investigated the ICL behaviour w.r.t. different pretraining data source and size, confirming that the source data properties can make or break ICL. Han et al. (2023) conducted an empirical study to show that challenging examples and long-tail tokens promote ICL by making the long-range information gain difficult. On the same lines, Chan et al. (2022) demonstrated that a large number of rarely occurring classes facilitate ICL. These works align with our generalized argument that a complex enough IWL task promotes ICL. Similarly, Razeghi et al. (2022) found a correlation between the term frequency of the input data in the pretraining corpus and ICL performance, further underscoring the influence of data characteristics. For local data patterns, studies by Olsson et al. (2022), and Chan et al. (2022) demonstrated that burstiness and repetitive structures within training sequences are vital for ICL. In a similar direction, Chen et al. (2024) have argued that parallel structures in the pretraining textual data, which follow a similar semantic or syntactic template, facilitate ICL in language models. They include repetitions in the parallel structures however their independent significance is not well disambiguated. Shi et al. (2024) studied the impact of combining the documents within the context window for language modeling and different textual in-context tasks. They showed a clear improvement in the ICL performance once the model was trained with a sequence of related documents showing the importance of data patterns in the pretraining. Our work studies local data patterns present in the training sequences, showing that simple copy-based data repetition, which can be seen as a special case of parallel structures is a major factor for stable ICL performance.

**Meta-learning vs ICL.** Meta-learning and In-context learning aim to enhance rapid adaptation to new tasks but operate through distinct mechanisms. Meta-learning often utilizes episodic pretraining where the model is pretrained with randomly sampled classes and label mappings in each episode, which enables fast adaptation to new unseen tasks during the testing phase with only a few parameters updates (Finn et al., 2017; Santoro et al., 2016; Snell et al., 2017). In this way, the model learns how to learn the task but does not learn the input label mapping directly due to the episodic permutations, which present an almost impossible learning task. On the other hand, there is no explicit pretraining objective for in-context learning. The input label mapping is crucial for the in-weight learning task and must not be harmed by the ICL abilities. Thus, while meta-learning explicitly optimizes for adaptability and robustness during training, ICL leverages pretraining knowledge like the original input-label mapping for adaption through information provided by input context. In this paper, we focus solely on concepts relevant to in-context learning but draw connections to the meta-learning pretraining via episodic pretraining and task difficulty when analyzing the choice of the in-weight learning task.

## 3 EXPERIMENTAL SETUP

We conducted an analysis to show how different properties impact ICL. We train an autoregressive model GPT-2 (Radford et al., 2019) on different sequences containing image-label pairs from image classification datasets widely popular in the FSL literature (Lake et al., 2015; Bertinetto et al., 2019; Fei-Fei et al., 2004; Cimpoi et al., 2014).

### 3.1 DATA SEQUENCING

The autoregressive model in this work is trained with a sequence length of $2L + 1$ with $L$ image-label pairs in the context followed by a query image, as shown in Figure 2. The in-weight learning objective is to predict the label of the last image, which is the $(2L+1)$-th token. Training sequences consist of a mixture of (1) *standard sequences*, in which sample-label pairs are randomly selected from the training dataset, and (2) *in-context sequences*, where the query image-label information is enforced to be present in the sequence by using a pair similar to the query image-label pair. The proportion of each sequence type in the total amount of training sequences is treated as a hyperparameter. Using in-context sequences, the model is implicitly regularized to attend to the similar context tokens without completely relying on the model weights.

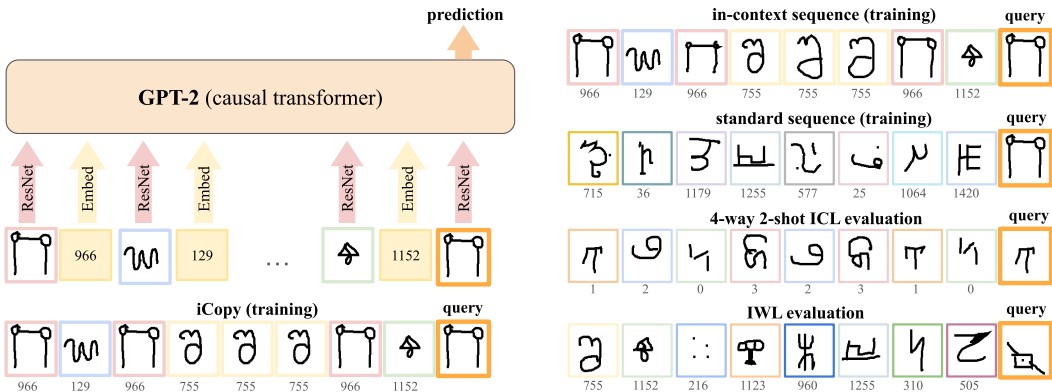

Figure 2: Illustration of our experimental setup. We use a causal transformer GPT-2 for sequential image-label pairs, treating each image or label as one token. During training, we alternate between in-context and standard sequences where we introduce a new form of in-context sequence *iCopy* - sequence with instance copy-pasting (repetition) sampling strategy (see Section 4 for more details). Evaluation includes few-shot classification (2-way 4-shot or 4-way 2-shots) on holdout classes for ICL and standard multi-class classification on validation split for IWL.

**Sequence notations.** Throughout the paper, we represent sequences using strings of $L$ characters, where each character corresponds to an image-label pair from a specific class. This is followed by

the query image token of class (Q). As an example, we denote sequences of length $L = 8$ as follows: we compare models trained with in-context sequences of different formats from (3xQ-3xA-B-C) to (Q-A-B-C-D-E-F-G), where (3xQ-3xA-B-C) means three image-label pairs are from the query(Q) class, three other image-label pairs from a non-query(A) class and other two image-label pairs are from two other non-query(B,C) classes. (Q-A-B-C-D-E-F-G) means only 1 sample is from the query class and the other 7 samples are from different non-query classes. The sequence order is shuffled during training. Standard sequences have the (A-B-C-D-E-F-G-H) format, where all image-label instances are from different non-query classes.

## 3.2 TRAINING AND EVALUATION DETAILS

We train a causal GPT-2 (Radford et al., 2019) transformer to predict the label $y_q$ of the query image $x_q$ given a sequence of L interleaved image-label pairs: $(x_1, y_1, x_2, y_2, ..., x_L, y_L, x_q, ?)$ as illustrated in Figure 2. Each image-label pair is converted into token embeddings separately. The model is trained to maximize the likelihood of the next token, with the loss applied to the final query output, thus using last-token prediction as the IWL training objective.

**Dataset construction.** We conduct our controlled experiments and analysis on the Omniglot dataset (Lake et al., 2015) and scale to more realistic visual datasets. Omniglot contains 1623 classes with 20 images each. Following previous work (Chan et al., 2022), we use 1600 classes for training and the remaining 23 as novel classes for ICL evaluation. More experiments (in Section 4) are performed on datasets including CIFAR-100 (Bertinetto et al., 2019), Caltech-101 (Fei-Fei et al., 2004), and DTD texture datasets (Cimpoi et al., 2014). ICL evaluation is performed using 20, 10, and 10 novel classes for CIFAR-100, Caltech-101, and DTD datasets, respectively. More experiment details are included in the Appendix B.

**Data sequencing details.** In all supervised image classification experiments, the models are trained with a mix of in-context and standard sequences. Models with instance discrimination (self-supervised learning) tasks are trained with 100% in-context sequences. ICL is evaluated in a few-shot classification setting for 2-way-4-shot and 4-way-2-shot tasks. We show results in the main paper for the more challenging 4-way-2-shot setting, while other results are included in the Appendix B.1 and Appendix C.3. This evaluation is performed on held-out novel classes. The trained classifier output is used for the few-shot evaluation using label mapping from 0-1 or 0-3 corresponding to both evaluation tasks. IWL is evaluated for the multi-class classification task on the held-out samples from the trained classes. The standard sequences with (A-B-C-D-E-F-G-H) format are used for IWL evaluation (see Figure 2). The same pre-set 10K and 3.2K sequences are used for ICL and IWL evaluation, respectively.

## 3.3 BASELINE

Our proposed baseline model is trained on the Omniglot dataset with a mix of standard and in-context sequences with 10% and 90% probability respectively. The standard sequences follow (A-B-C-D-E-F-G-H) format, and in-context sequences have a high burstiness with (3xQ-3xA-B-C) format. The baseline model with in-context sequence format (3xQ-3xA-B-C), achieves strong ICL and IWL performance (shown in Figure 3). Prior work observed poor IWL performance with strong ICL performance. We think this happens because they utilize standard sequences in (Q-A-B-C-D-E-F-G) format with at least one repetition of the query in context, which undermines the in-weight learning task. However, similar to prior work, we observe that the ICL performance is transient as training proceeds for this baseline model.

## 4 WHAT PROMOTES IN-CONTEXT SOLVING ABILITY?

The term "in-context solving ability" or "look-up mechanism" refers to the ability of the transformer-based model, where the model can prioritize the usage of information present within its current input context to generate responses. The model is not explicitly trained to learn this mechanism, however, it is crucial for the model to solve the task based on the in-context information. This work examines a previously proposed in-context sequence strategy, which are motivated by the trends observed

in the pretraining corpora of LLMs and further analyzes the impact of repetitions in training data sequences.

**Burstiness.** Burstiness is an inherent feature of natural sequential data. Chan et al. (2022) showed that the burstiness property in training sequences is crucial for ICL and demonstrates its effectiveness. Burstiness, in a classification setup like ours, is defined as the number of occurrence of samples from the same class as the query sample in the context. Models with this strategy are trained with a mix of highly bursty (`3xQ-3xA-B-C`) and non-bursty (`A-B-C-D-E-F-G-H`) sequences to obtain both ICL and IWL. High-bursty and non-bursty sequences are types of in-context and standard sequences, respectively.

**Repetitions.** Although burstiness is typically a natural feature of sequential data, repetitions are also a specific aspect of burstiness. As motivated in the introduction, the analysis of pretraining corpora reveals frequent repetitions of n-grams in different pretraining corpora, indicating that pretraining text exhibits burstiness not only via clustered phrases of synonyms or similar topics but also through exact repetitions of different lengths. Therefore, we conducted a study to examine the impact of repetitions on ICL. We find that simply *copying* the query-label pair into the input sequence during training develops a strong look-up mechanism.

**iCopy.** In light of previously introduced ways of fostering stronger in-context learning, we refer to the sequences with conceptual repetitions for image classification tasks as iCopy. iCopy utilizes an instance-based copying mechanism where the query-label pair is copy-pasted within the sequence of defined burstiness. This simplifies the matching process between a query token and its duplicates across different token positions in the context. Since the positions of the copied token are shuffled and sequences with repetitions are mixed with standard sequences, the model learns a generalized look-up mechanism across the whole context. The sequences with repetitions can also be considered as a special case of an in-context sequence with low-burstiness, denoted as (`Q-A-B-C-D-E-F-G iCopy`) in this work. This copied version in the context can be an exact copy or an augmented version of the original query sample-label pair.

In the previously proposed high-burstiness strategy, samples from the same class as the query appear multiple times, whereas the sequences with repetitions may only contain one occurrence of the same instance as the query sample. However, repetitions show enhanced results when combined with the high-burstiness strategy (`3xQ-3xA-B-C iCopy`), comprising multiple instance copies in the context.

We conducted ablations and analysis to study how repetitions perform against previously introduced data distributional properties on ICL and IWL tasks.

### 4.1 iCopy promotes look-up

**Repetitions are sufficient for ICL.** We compare in-context sequences with high-burstiness (`3xQ-3xA-B-C`) and sequences with one repetition (`Q-A-B-C-D-E-F-G iCopy`), observing similar ICL peak performance for both types of in-context sequences but with reduced transiency for the sequences with repetition. Combining the repetitions and high-burstiness strategies (`3xQ-3xA-B-C iCopy`) further reduces ICL transiency. Using low-burstiness without iCopy (`Q-A-B-C-D-E-F-G`) shows no ICL ability. ICL and IWL performance curves are shown in Figure 3 a,b, where we can also observe that the repetitions in iCopy sequences do not harm the IWL performance.

**Skewness is not necessary.** We compare models trained with skewed and uniform label distribution, using 7200 samples randomly selected from 992 classes. The sequences with repetitions and uniform distribution outperform the high-burstiness strategy with Zipfian distribution (Figure 3c). Thus showing that a skewed distribution improves ICL over the baseline, it is not necessary for ICL.

### 4.2 iCopy reduces ICL transiency

We observe that the ICL performance is transient across all look-up strategies studied in this work (see Figure 3), a behavior also observed by prior work (Singh et al., 2023). We observe using iCopy results in reduced ICL transiency as shown in Figure 3 (a). Previous work (Singh et al., 2023) speculate that the competition between IWL and ICL circuits is responsible for the transient

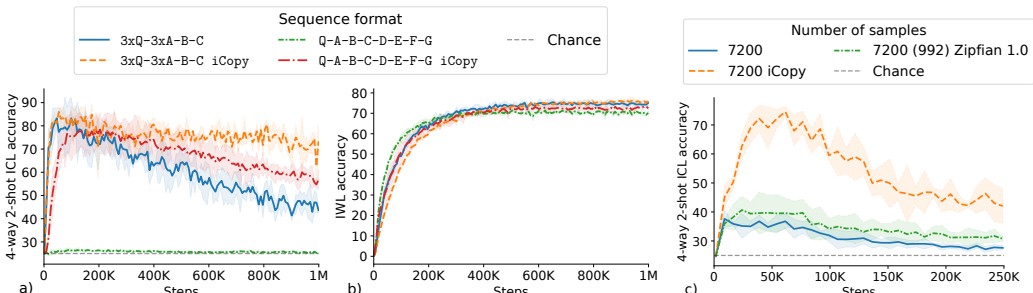

Figure 3: a), b): Effect of the repetitions on ICL performance. Even in the low-burstiness scenario, sequences with repetitions (Q-A-B-C-D-E-F-G iCopy) show strong ICL and improved stability compared to sequences with high-burstiness (3xQ-3xA-B-C). When in the high-burstiness scenario, sequences with repetitions (3xQ-3xA-B-C iCopy) further improve ICL stability while not harming IWL performance. c) Skewness is not necessary for ICL. The sequences with repetitions (3xQ-3xA-B-C iCopy) strategy achieves better ICL performance compared to using skewed Zipfian distribution. The number of samples (7200) and classes (992) are kept the same for all cases.

behavior of ICL. We think this likely happens because, as the training progresses, the IWL task becomes easier and can be solved purely using in-weight knowledge, thus weakening the in-context look-up mechanism. To support this argument, we show that by using a harder IWL task of instance discrimination with iCopy, transiency is nearly eliminated (Figure 4a). In an instance discrimination task, each sample is treated as a separate class, also referred to as a self-supervised learning objective in the literature. This might also explain why LLMs can retain IWL and ICL performance even after training: language modeling is a complex IWL task due to the complexity of natural language, including ambiguity, long-range dependencies, and large vocabulary. More on this in Section 5.

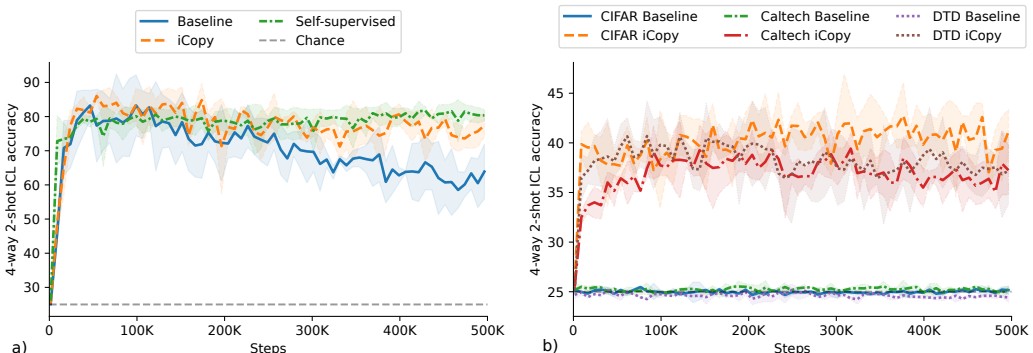

Figure 4: a) Repetitions reduce transiency. When considering high burstiness scenarios, sequences with repetitions (3xQ-3xA-B-C iCopy) reduce ICL transiency compared to the baseline (3xQ-3xA-B-C). The instance discrimination IWL task with high burstiness and repetitions eliminates ICL transiency b) Scaling experiments on Cifar-100, Caltech-101, and DTD. Our best model setting with repetitions and high-burstiness strategy shows strong 4-way-2-shot ICL performance on all three datasets.

### 4.3 iCopy PROMOTES THE INDUCTION HEADS

Prior works show that the formation of an induction head leads to in-context learning (Olsson et al., 2022; Reddy, 2024). Induction heads are a result of the two-layer circuit that performs match and copy operations from the context. For a bigram sequence containing image-label pairs, first, the label tokens attend to the previous image tokens and copy the information into their representation. Then the query image token attends to the matching label tokens and copies the label information.

Since the repetitions accompanied by high burstiness create a strong look-up mechanism, we can obtain ICL with a GPT-2 model using only 3-layer with 1-head per layer. We observe that it is not possible to obtain a clear ICL performance with the same architecture using only the high-burstiness strategy.

The simplified model allows us to analyse the underlying attention mechanism in each head independently and track the induction head formation throughout the training. As shown in Figure 5(c, d), we can observe an induction circuit formation using the combination of repetitions and high-burstiness strategy, where layer-1 (L1) consists of previous token attention heads (Fig. 5 (c)) and layer-2 (L2) shows query image to matching label head attention (Fig. 5 (d)). The snapshot represents the peak of ICL performance for the model with repetitions. We do not show Layer-0 (L0) attention map since it does not show noticeable patterns. We do not observe any ICL performance and induction circuits in the model using only high-burstiness (Fig. 5(a,b)) at this snapshot. We also track progress metrics like label-image attention to measure previous token attention in layer-1 of the induction circuit, image-label attention to measure query image to label attention in the layer-2 of the induction circuit, and image-image/label-label attention to measure the strength of token representation (included in the Appendix C.3). We further analyse the induction head formation using a low-burstiness setup with repetitions in the Appendix C.3.1, where we also observe induction head formation.

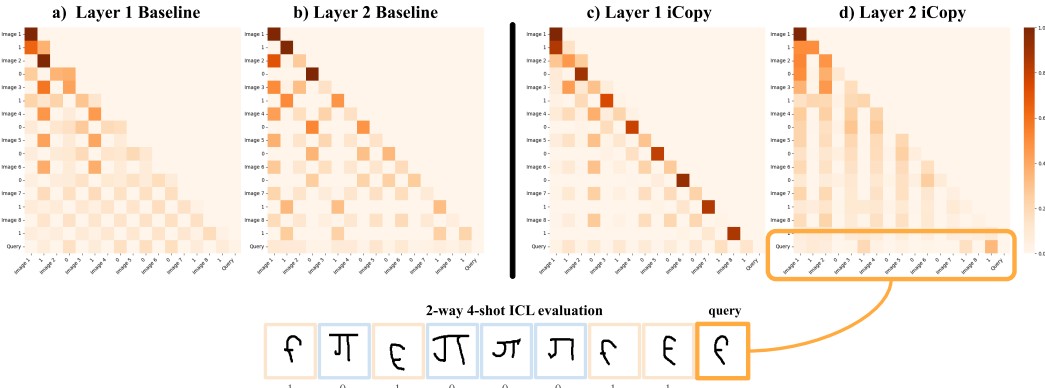

Figure 5: Induction head analysis of GPT-2 model with three layers and one head on sequences with only high burstiness (baseline) and repetitions with high burstiness(iCopy). We compare the attention maps of layer-1 (L1) and layer-2 (L2) for the baseline and model with repetitions. The attention maps correspond to the 2-way-4-shot evaluation sequence shown at the bottom. Layer-0 (L0) is not shown because it has no noticeable patterns. High-burstiness, when combined with repetitions, shows clear formation of the induction head: L1 learns the label-image pair associations, while L2 performs association to the correct label. In contrast, no induction head formation is seen in the baseline model.

### 4.4 iCopy promotes ICL on other datasets

Repetitions in combination with high-burstiness scales well to more realistic datasets like CIFAR-100, Caltech-101, and DTD. We observed strong ICL performance on all three datasets. Using only high burstiness in the in-context sequences (Baseline) does not show any in-context learning for these datasets as shown in Figure 4b.

## 5 Does IWL objective matter for ICL?

As described in Section 4.2, we believe the look-up mechanism on its own is not sufficient for a stable ICL performance and a choice of the appropriate IWL objective plays an important role. In particular, we hypothesize that if the in-weight task is too simplistic, the look-up mechanism does not emerge or results in a transient ICL performance since the model can optimize for the IWL objective without needing to attend to the context. Therefore, the in-weight task must have a minimum level

of complexity to give rise to the look-up mechanism. A related study (Chan et al., 2022) showed that a long-tail distribution and an increase in the number of classes improve ICL, connecting these properties to natural language data. In this work, we additionally provide an explanation of why these properties enhance ICL - by increasing the complexity of the IWL task.

We propose four different ways of regulating the IWL task difficulty - changing the number of training classes, changing the number of samples used for training, training with noisy labels, and switching to the instance discrimination task. Here, we show how each of the proposed techniques influences the ICL and IWL performance.

**Number of classes:** We observe that increasing the number of classes monotonically improves ICL performance, as illustrated in Figure 6a. This finding follows similar insights in prior work (Chan et al., 2022; Reddy, 2024). However, these works explain the improved ICL capabilities due to the large number of rarely occurring classes. We interpret it as just one out of many ways to make the IWL task harder. Please refer to Appendix D for more details.

**Skewed distribution:** Keeping the total number of samples the same, we compare the ICL performance of a model trained with balanced and imbalanced (Zipfian) distributions. We observe improved ICL performance with skewed distribution, as illustrated in Figure 6b. This experiment is a confirmation of prior work (Chan et al., 2022), which also shows improved ICL with the increased long-tail distribution. Imbalancing the label distribution is a known way to make the in-weight learning task harder. Please refer to Appendix D for more details.

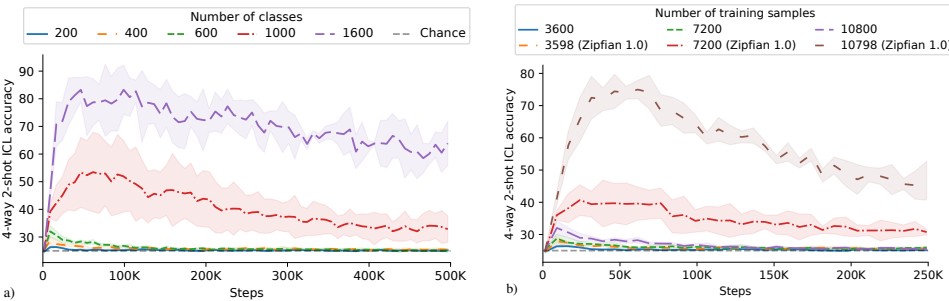

Figure 6: a) *Number of classes.* When increasing the number of classes form 200 to 1600, we observe a clear improvement in the ICL performance as the IWL objective is now harder. b) *Skewed distribution.* By having the same number of training samples, we report better ICL performance with the skewed data distribution.

**Learning with noisy labels:** We increase the complexity of the IWL task by adding label noise, where labels of certain samples in the sequence are randomly assigned to another training class. Label noise is only applied to the standard sequences. We train a supervised model with 600 classes with three increasing levels of label noise percentage. Figure 7 (a, b) shows that as the label-noise ratio increases, the ICL performance improves while the IWL performance reduces, showing how the task is harder with more noise.

**Instance discrimination task:** Using the copy-based look-up strategy with repetitions in the sequence, we devise a more complex IWL task by moving from supervised to self-supervised learning. We design a task based on the instance discrimination (Wu et al., 2018) objective, where the model is trained to classify each sample as its class. [1] We train the baseline model in the supervised high burstiness setting with 3600 samples from 200 classes where the ICL does not emerge, illustrated in Figure 7 c. Whereas, when we train with the instance discrimination objective on the same number of samples, we obtain very strong and stable ICL performance, which is the result of hard IWL task and strong in-context look-up.

---

[1] This would mean that in the case of the Omniglot instance discrimination setting with 1600 classes and 18 exemplars per class, the total number of classes would be 28800.

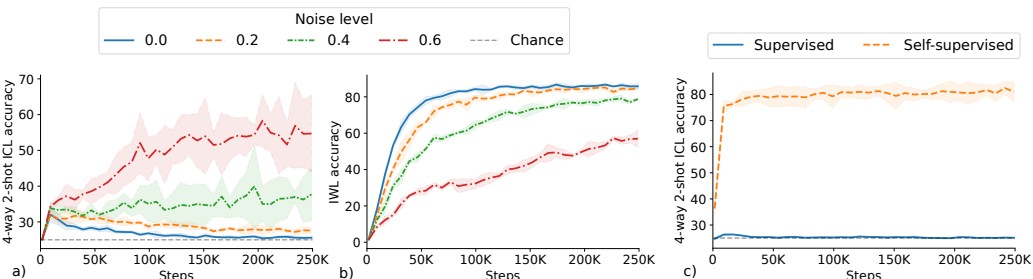

Figure 7: a, b) Effect of label noise on ICL and IWL performance. The model trained with a high label noise percentage achieves high ICL accuracy since the IWL task becomes harder, which can be seen by the drop in IWL accuracy. c) The instance discrimination IWL task shows strong and stable ICL performance. Both models are trained on 200 classes comprising 3600 samples. The baseline supervised model does not show ICL as it is a much simpler IWL task.

# 6 DISCUSSION

**Key insights.** We identify that the balance between training data that provides opportunities to use the in-context look-up mechanism and the complexity of the model training objective influences the emergence and stability of ICL. We show that the conceptual repetitions, which are also commonly present in textual data sequences, induce a strong in-context look-up mechanism. Repetitions combined with high-burstiness in the training sequences result in peak ICL performance. Models with such in-context look-up mechanisms ensure strong ICL performance with no harm to the IWL component, while also demonstrating reduced transiency in ICL performance. This supports our hypothesis that the conceptual repetitions in the data are strong drivers for the ICL ability. As shown in Section 5, training with a complex in-weight learning objective leads to improved and non-transient ICL performance. This conclusion aligns with the complex training objective and observed nature of ICL performance in LLMs.

**Limitations.** Although the ranking between different compared methods is clear, we observed a large variance in the ICL performance curves *w.r.t.* random seeds where the IWL task is simple. We believe one of the reasons for this is the sensitivity of ICL towards training sequences (Press et al., 2023). While our approach transfers well to other visual datasets and different training objectives, showing a similar analysis on real-world sequential data is out of scope of this work. We also found that ICL is sensitive to certain model design choices such as model weight initialization and image embedding architecture. For instance, using a truncated normal distribution for initialization improves the stability of ICL performance across different seeds. Addressing the robustness of ICL is a goal for future work.

**Broader implications.** Models with in-context learning ability allow users to leverage large models for various downstream tasks without the need to adapt the model weights by simply defining the task during inference. We provide an analysis of the training sequences showing that conceptual repetitions are influential and they reduce the need for specific properties in the data distribution, giving more flexibility to obtain ICL for different domains and applications. From a research perspective, our findings allow for analyzing internal workings in a controlled setup, facilitating a better understanding of ICL. Our work is based on mainly synthetic training sequences and thus has no direct societal impact.

**Future work.** We believe that our findings about the essential properties of the data sequence and design of in-weight learning objectives could be crucial for future large model training, where in-context solving ability is important. Our findings are agnostic of the data domain and should be applicable to LLMs, VLMs or other modalities.

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

## A    MODEL DETAILS

**Architecture details.** We train the GPT-2 model Radford et al. (2019) with 12 layers and 8 heads with an embedding dimension of 64. We use a smaller model for the induction head analysis experiments with 3 layers, a single head, and an embedding dimension of 64. GPT-2 expects a sequence-like format with aligned embedding size so we transformed our image-label pairs into separate image and label tokens, using a ResNet-like embedder for images and an embedding layer for labels. We initialized the model with a truncated normal distribution, which is important for training stability. We use a 3-block ResNet model  He et al. (2016) as the image embedder with output channel dimensions [64, 128, 256]. After that, a projection layer is added to match the embedding dimension of 64. We train the image embedding model and GPT model together from scratch. We notice that the emergence of ICL is sensitive to the input image embedder architecture.  We also report that pretrained embedders result in fast convergence of in-weight task and ICL failure cases, as shown in Figure 8.

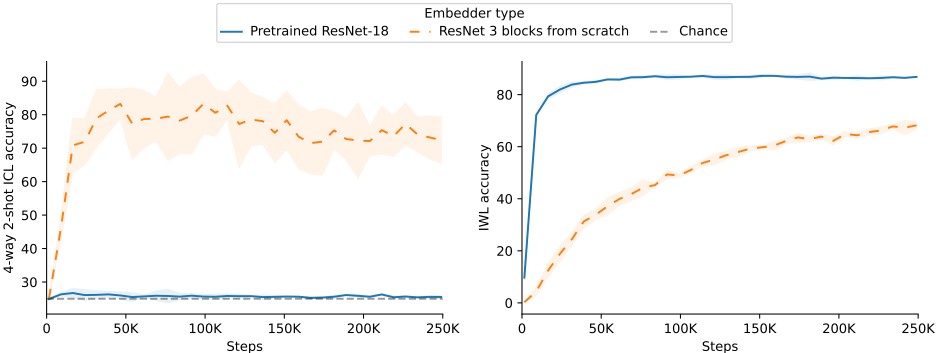

Figure 8: 4-way 2-shot ICL and IWL accuracy for different versions of the image embedder. Using pretrained models makes the in-weight task easier and ICL does not emerge.

**Hyperparameters.** We trained the model for different numbers of steps varying from 250k to 2M iterations using optimizer Adam  Kingma & Ba (2014) with betas (0.9, 0.99) and epsilon 1e-08. We use learning rate warm-up for 15K iterations with a square root decay scheduler with a maximum learning rate value of 6e-4. We find that ICL performance is enhanced with longer warm-up periods. We perform gradient clipping to value 1.0. We trained the model with a batch size of 16 on a single Nvidia RTX 3090 where 500k iterations took around 12 hours.  For all experiments, we run the approach for 3 random seeds, except for label-noise experiments in Section 5, where we report performance over 5 random seeds.

## B    EXPERIMENT SETUP DETAILS

**Training details.** The model is trained with a mix of two types of sequences with different burstiness forms: in-context sequences and standard sequences. In-context sequences can have multiple reoccurring samples from the same class as the query, whereas standard sequences have all samples selected from random classes.  The high-burstiness strategy uses in-context sequences with three re-occurrences and iCopy strategy uses only one re-occurrence.  All the supervised models in this work are trained with a burstiness probability of 90%, which means 90% in-context sequences and 10% standard sequences. All self-supervised models use 100% in-context sequences.

**Evaluation details.** We evaluate separately for both ICL and IWL. ICL evaluation is performed in a few-shot manner with 2-way 4-shots and 4-way 2-shots tasks. The evaluation is performed using the pretrained softmax classifier without any model update. We use label mappings 0 to $k$ with $k$ being the number of classes in the few-shot setting.  We notice that the ICL results are agnostic to label mappings. ICL is always performed on the hold-out classes not seen during training. ICL is performed over 10K presampled sequences to ensure a fair comparison. IWL is evaluated on the hold-out samples from the training classes where the sequence format must be (A-B-C-D-E-F-G-H) so that the model does not perform look-up, but relies only on the model weights.

## B.1 DATASETS

All analysis experiments are performed on the Omniglot dataset Lake et al. (2015). We further show ICL results on other visual datasets Cifar-100 Bertinetto et al. (2019), Caltech-101 Fei-Fei et al. (2004), and DTD textures Cimpoi et al. (2014) which are often used for benchmarking few-shot learning classification task. A few details about these datasets are included below:

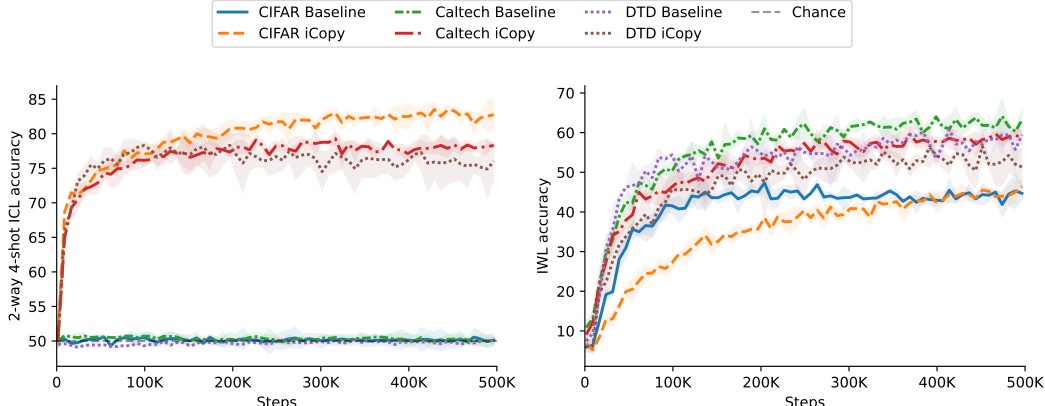

Figure 9: Scalling results on datasets CIFAR, Caltech and DTD shown for 2-way 4-shot ICL and IWL accuracies. Our iCopy reports strong ICL emergence and stability for the 2-way 4-shot scenario.

**Omniglot** consists of 1623 handwritten characters from 50 alphabets with 20 exemplars for each character. Unless stated otherwise, we use 1600 classes as the base classes and the remaining 23 classes (sampled from the official evaluation subset with seed 42) for the ICL evaluation. We create a train-validation split as 18-2. During the supervised training, we apply no augmentations except for resizing to 64x64. However, self-supervised setup benefits from mild augmentations (random crop resize to 64x64 with scale (0.5, 1.5) and horizontal flip). For the self-supervised experiments, we used a batch size of 216 and a learning rate of 1e-3.

**CIFAR-100** is a natural dataset consisting of 60000, 32x32 colored images divided into 100 categories with 600 examples from each one. We use 80 classes for supervised training and 20 classes for the ICL evaluation as it is given by the Cifar-100FS (Few-Shot) version of the dataset. We used 10% of the data for the validation. We do not apply any augmentations, but we resize the image to 64x64 for training and evaluation.

**Caltech-101** is a natural, imbalanced dataset with 101 classes with 40-800 images per class while most classes have about 50 images and each image is roughly 300x200 pixels. We randomly select 91 classes for the supervised training and the remaining 10 classes are used for ICL evaluation. During training, we use random resized cropping to 64x64 with scaling from 0.5 to 1.5, horizontal flipping and random rotation of 15 degrees.

**DTD** is a texture dataset consisting of 5640 images across 47 classes with 120 images from each class with a size ranging from 300x300 to 640x640. We use 37 classes for supervised training and 10 classes for the ICL evaluation and create a train and validation split with roughly 10% of data used for validation. We report better and more stable results with an image size of 128x128 and random resize with scale (0.5, 1.5).

**Results on CIFAR, Caltech and DTD.** The 4-way-2-shot ICL results are included in the main paper. Here, we show 2-way 4-shot and IWL performance for other datasets in Figure 9. We observe strong ICL performance using the combined strategy of iCopy and high-burstiness while the baselines with just high burstiness sequences do not show any ICL.

864
865
866
867
868
869
870
871
872
873
874
875
876
877
878
879
880
881
882
883
884
885
886
887
888
889
890
891
892
893
894
895
896
897
898
899
900
901
902
903
904
905
906
907
908
909
910
911
912
913
914
915
916
917

## C  IN-CONTEXT LOOK-UP MECHANISM

In this section, we report more results and details on different in-context look-up strategies. Here, we also try to answer a question "Can burstiness be relaxed once the look-up mechanism has been formed?" To answer this question, we introduce burstiness scheduler, a training regime with which the burstiness is relayed from high-burstiness to low and the ICL is still maintained. We further provide induction head analysis with progress metrics for burstiness scheduling and iCopy strategy.

### C.1  BURSTINESS

First, we confirm previous findings about the importance of burstiness format in the training data. Figure 10 shows 3 different levels of burstiness from high burstiness level (`3xQ-3xA-B-C`) to low burstiness level (`Q-A-B-C-D-E-F-G`). The Figure shows that a larger magnitude of burstiness promotes better ICL performance. We report 4-way 2-shot, 2-way 4-shot ICL, and IWL performance.

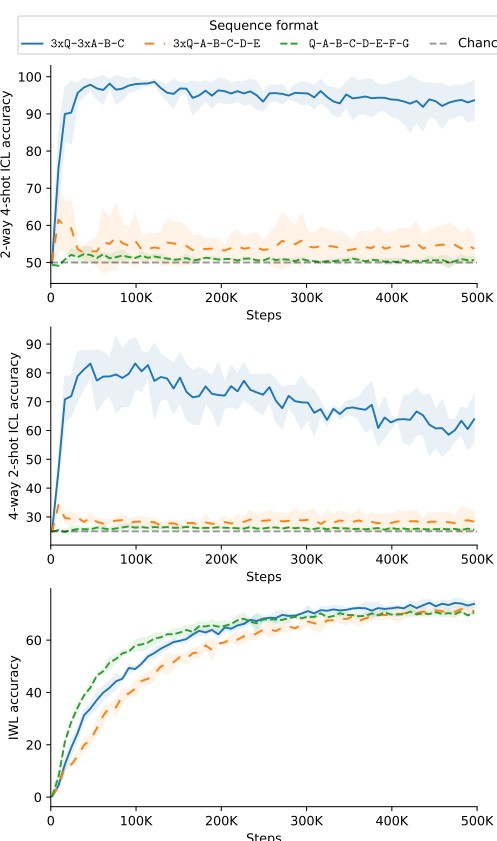

Figure 10: *Burstiness:* ICL and IWL accuracy for different burstiness format. High burstiness improves ICL performance without affecting IWL performance. 2-way 4-shot evaluation is easier ICL setup which results in more stable and less noisy performance than 4-way 2-shot. IWL performance remains same for all sequence formats.

### C.2  BURSTINESS SCHEDULING

Prior work Olsson et al. (2022); Reddy (2024) has observed that ICL tends to emerge abruptly due to a phase transition and is attributed to the formation of an induction head circuit. Motivated by these findings, we explore whether the magnitude of burstiness can be reduced in the in-context sequences once ICL is established since having consistent high burstiness in the training data is unnatural. We discover that the high burstiness requirement can indeed be gradually relaxed using a scheduler over the training process without harming the ICL performance. We also notice that

implementing burstiness scheduling creates a learning curriculum that enhances the robustness of induction heads. This, in turn, also improves ICL performance.

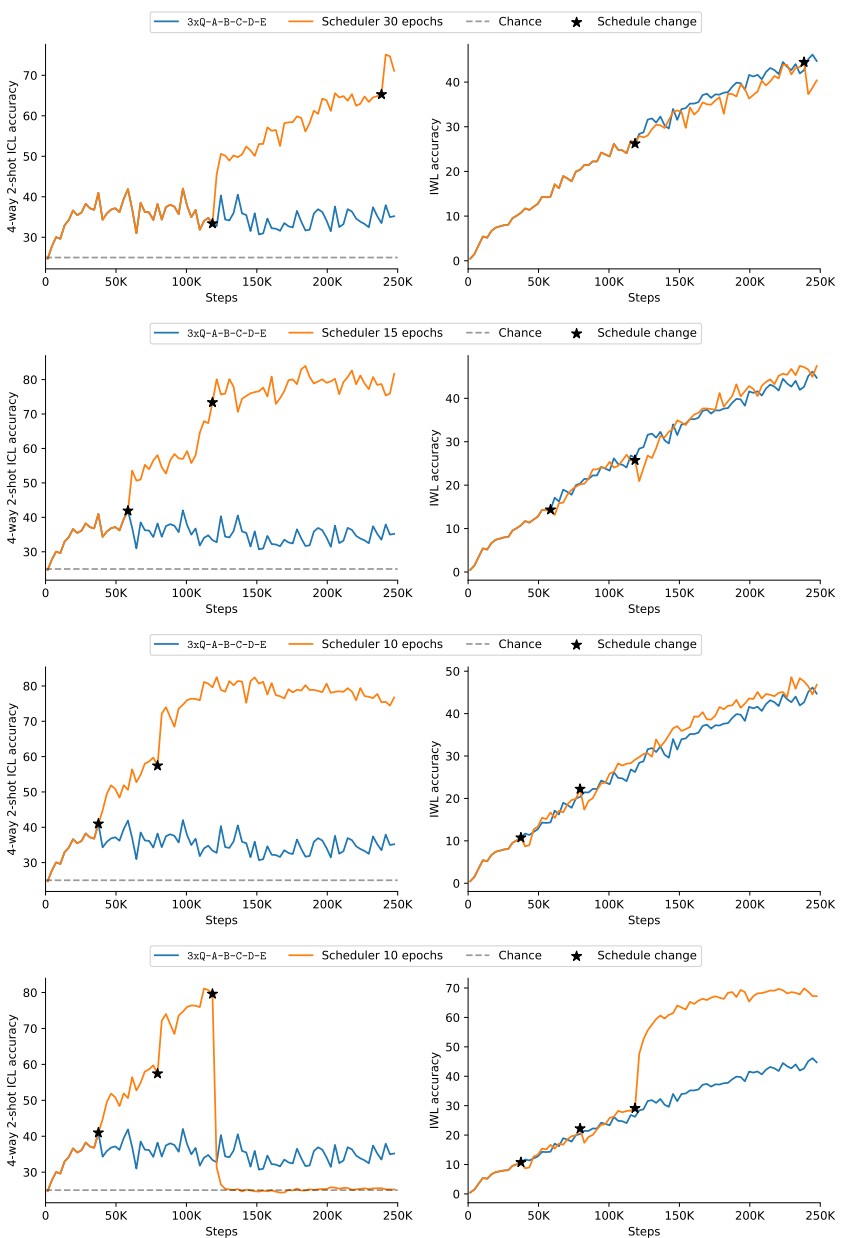

Figure 11: Results of different burstiness schedulers. First row has later change in burstiness compared to the second and third row. We observe early change reaches peak ICL performance faster and remains stable. The last row shows, if the burstiness is completely removed from the sequences, ICL becomes non-existent as the model is now trained with only standard sequences which promotes IWL performance.

We constructed different burstiness schedulers where we relaxed the burstiness forms at different points during the training process, as shown in Figure 11. We start model training with a high burstiness of 3 (`3xQ-3xA-B-C`) and then relax gradually to a low burstiness of 1 (`Q-A-B-C-D-E-F-G`). We find that it is better to activate the scheduler in the early phases of training, but only after the initial emergence of ICL as shown in Figure 11. We show that the burstiness forms can be relaxed over time, but we still need a minimal level of burstiness to maintain ICL performance. The last row on

Figure 11 shows a sudden drop in ICL performance when burstiness was completely removed from the sequence format ((`Q-A-B-C-D-E-F-G`) to (`A-B-C-D-E-F-G-H`)).

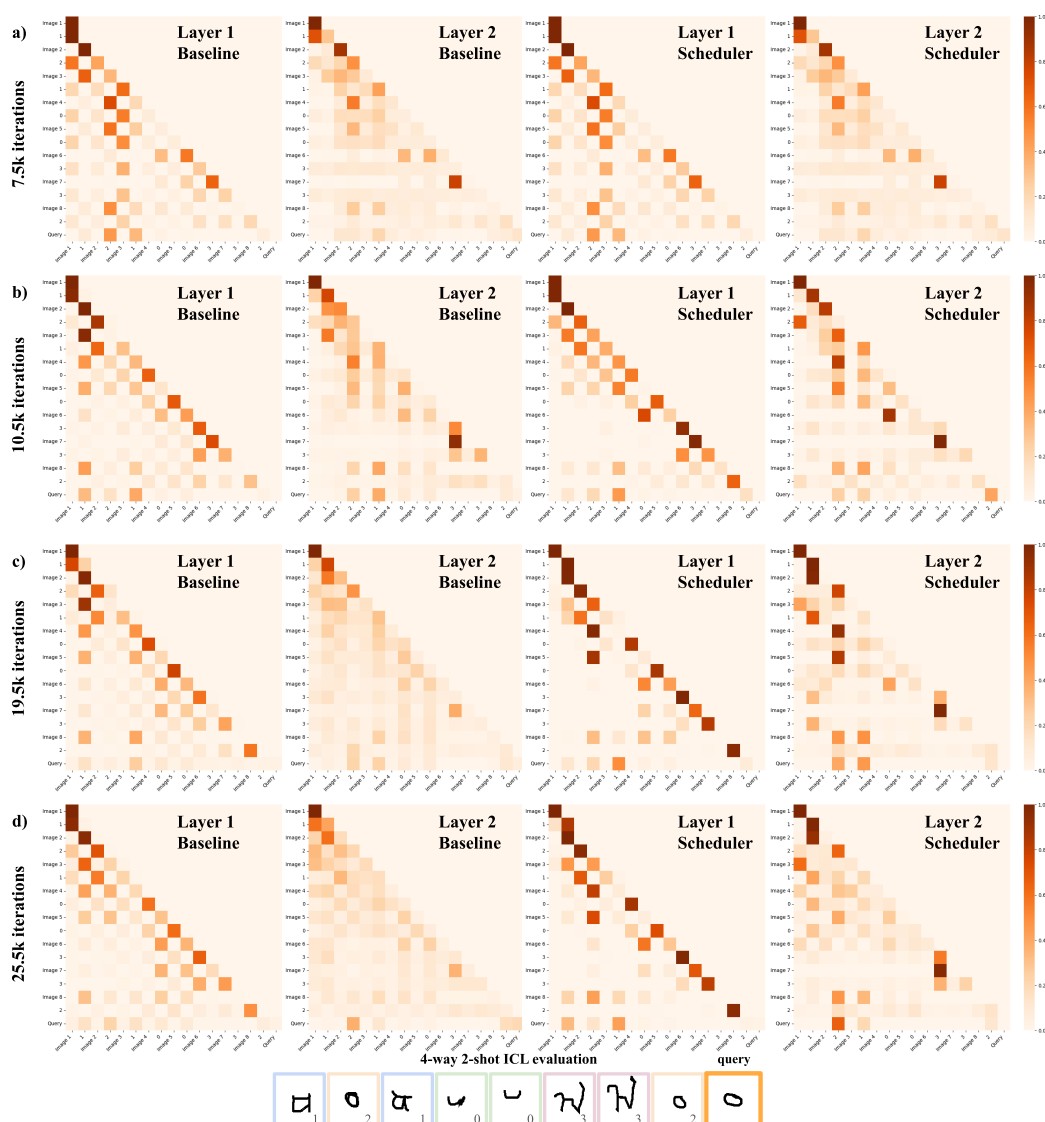

Figure 12: Induction head analysis of GPT-2 model with 3 layers and 1 head for the burstiness scheduling strategy. We compare the attention maps of layer-2 (L1) and layer-3 (L2) for the baseline (columns 1 and 2) and model with burstiness scheduler(columns 3 and 4) for one 4-way 2-shot sequence shown in the last row. Rows represent different phases throughout the training - row a) is before the scheduling activation (∼7.5k iterations), row b) is after the activation (∼10.5k iterations), row c) is at the peak of ICL performance for the baseline (∼20k iterations), row d) is at the peak of ICL performance for the scheduled model (∼25k iterations).

Similar to the analysis shown in Section 4.3, we show induction head 2-layer subcircuit analysis at different stages of the scheduler. We conduct the analysis using a small GPT-2 model with 3 layers, 1 head, and an embedding dimension of 64 with the scheduling formats from (`3xQ-3xA-B-C`) to (`Q-A-B-C-D-E-F-G`). We show attention maps for second (layer-1) and third (layer-2) layers. We do not show attention maps for the first layer (layer-0) since it does not have noticeable patterns. Figure 12 shows the effect of using a burstiness scheduler at different stages of the training process. We observe that the induction head becomes stronger with each change in the burstiness level. In the last row, when the ICL reaches its peak performance using the burstiness scheduler, we can see

a strong presence of induction heads in Layer-2. Different checkpoints(rows) in the Figure 12 can be connected to performance curves in Figure 13.

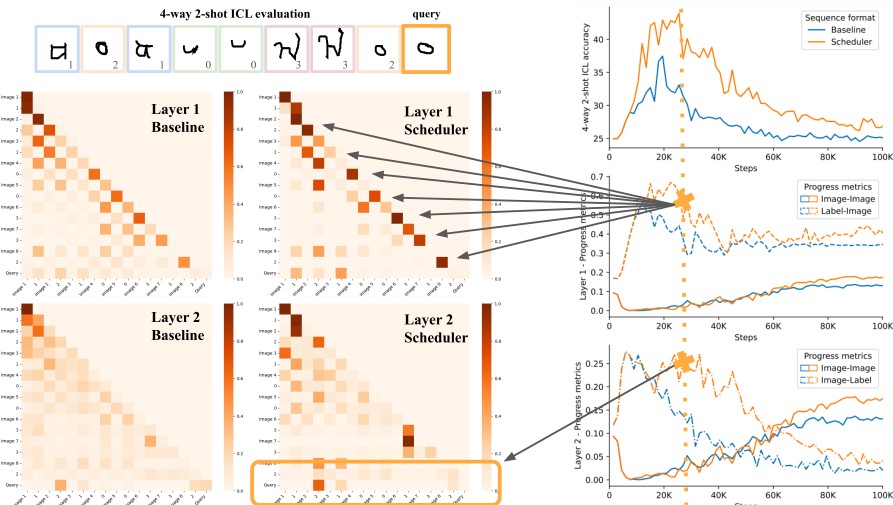

Figure 13: Induction head analysis of the GPT-2 model with 3 layers and 1 head for the baseline (3xQ-A-B-C-D-E) and burstiness scheduling ((3xQ-3xA-B) → (Q-A-B-C-D-E-F-G)) *Left:* Figure shows attention maps for second (L1) and third (L2) layers. *Right:* Progress metrics for the baseline and burstiness scheduling are shown. The label-image attention from the first layer circuit and image-label attention for query image in the second layer circuit are indicated with the arrows.

**Progress metrics:** We study the formation of induction heads and the emergence of ICL using three progress metrics: image-image, label-image and image-label. The image-image measures the average attention between all image tokens to all other image tokens. The label-image measures the average attention between each label token to its previous image token. The image-label measures the average attention between the query image and the correct label token.

Figure 13 compares attention maps of Layer-1 and 2, when the ICL reaches peak performance using the burstiness scheduler. We observe strong attention from query image to correct label from the context, which is also considered as induction head formation. Layer-1 performs label to previous image token mapping when ICL emerges. This can be seen in the burstiness scheduler case, where label-image similarity also peaks when ICL peaks, as shown in the middle row curves. Layer 2 performs query image to correct label mapping when ICL emerges. This can be seen in the burstiness scheduler case, where image-label similarity peaks when ICL peaks, as shown in the last row curves. We do not observe any dominant patterns for the baseline model without burstiness scheduler. Baseline usually only does image-image or label-label mapping.

## C.3 ICOPY: DETAILS

Our iCopy strategy shows good results and stability on both easier and harder ICL tasks while also reporting reduced transiency. On the easier 2-way 4-shot ICL task, our iCopy strategy proves to be highly stable and with small variance.

In Section 4.1 we show how iCopy does not necessarily need high burstiness in the training sequences or skewness in data. Here, we give more results on those experiments and report 2-way 4-shot ICL accuracy (easier setup), shown in Figure 14. We also show 2-way 4-shot ICL performance for models trained with Zipfian data distribution without iCopy and uniform data distribution with iCopy. We observe that iCopy is sufficient to obtain ICL and skewness is not necessary to obtain ICL, shown in Figure 15.

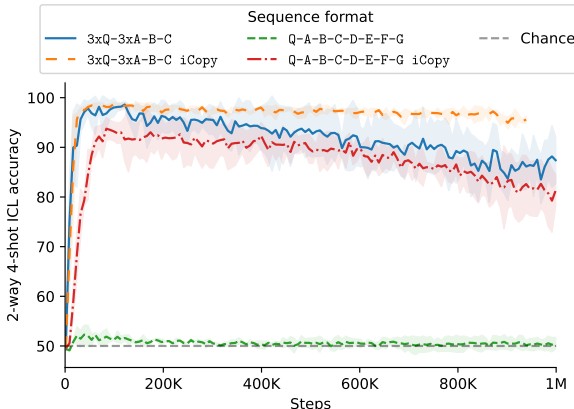

Figure 14: *Repetitions are sufficient.* Effect of our iCopy strategy on 2-way 4-shot performance. Our iCopy with low burstiness shows stable and strong ICL ability. When paired with high burstiness, performance and stability improves even more.

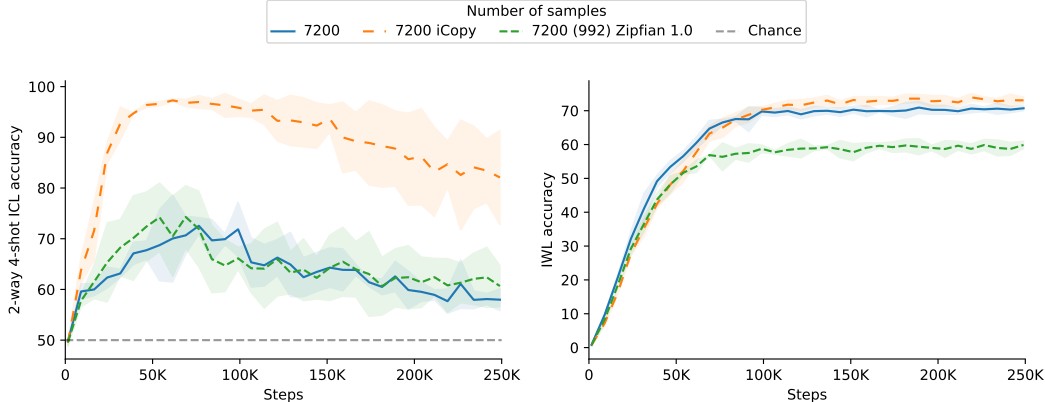

Figure 15: *Skewness:* We fix the number of samples to 7200 and compare the models trained with skewed data distribution without iCopy and uniform distribution with iCopy. The skewed distribution shows improvement over uniform sampling, but our iCopy with uniform sampling performs significantly better.

### C.3.1  ANALYSIS

Here, we show the formation of induction heads using only iCopy strategy with in-context sequence format (`Q-A-B-C-D-E-F-G iCopy`), shown in Figure 16 The figure also shows progress metrics using only iCopy strategy.

**Progress metrics.** We study the formation of induction heads and the emergence of ICL using three progress metrics: image-image, label-image and image-label. The image-image measures the average attention between all image tokens to all other image tokens. The label-image measures the average attention between each label token to its previous image token. The image-label measures the average attention between the query image and correct label token.

All the experiments were conducted with GPT-2 model with 3-layers and 1-head. We observe strong ICL performance; however, it is transient in nature. In Layer-1, label tokens show strong attention towards previous image tokens. This is also indicated by the label-image progress metric in the middle row curves, which peaks concurrently with ICL performance. In Layer-2, query image tokens attend strongly to the labels of the correct images. This is also indicated by the peaked image-label measure, which correlates with peak ICL performance. We observe as the image-label measure in Layer-2 becomes transient, the ICL performance also becomes transient. We do not observe

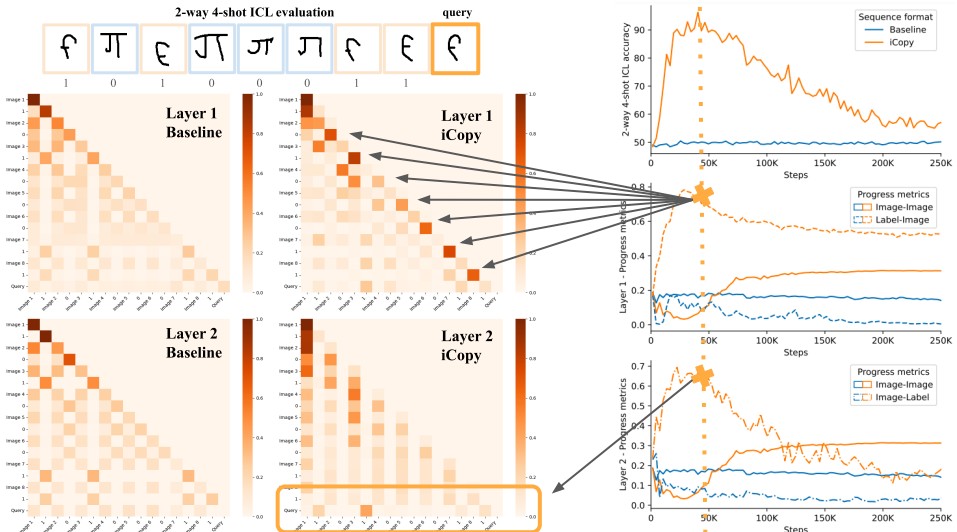

Figure 16: Induction head analysis of the GPT-2 model with 3 layers and 1 head for the baseline (`Q-A-B-C-D-E-F-G`) and iCopy (`Q-A-B-C-D-E-F-G iCopy`) with both low burstiness accompanied by the progress metrics for layer-2 (L1) and layer-3 (L2). We show the label-image and image-label attentions are closely related to the ICL emergence and follow the similar trend as ICL performance.

any dominant patterns for the baseline model using in-context sequence as (`Q-A-B-C-D-E-F-G`). Baseline only does image-image or label-label mapping in both layers.

## D  ROLE OF IWL TASK

As described earlier, we believe the look-up mechanism on its own is not sufficient for a stable ICL performance and a choice of the appropriate IWL task plays an important role. We proposed 4 different ways of regulating the IWL task difficulty - changing the number of training classes, changing the number of samples used for training, training with noisy labels, and switching to the self-supervised training regime. Here, we show how each of the proposed techniques influences the ICL and IWL performance.

**Number of classes**    As already mentioned before, we observe that increasing the number of classes monotonically improves ICL performance. This finding follows similar insights in prior work (Chan et al., 2022; Reddy, 2024). However, these works explain the improved ICL capabilities by the large number of rarely occurring classes. We interpret it as just one out of many ways to make the IWL task harder.

To simulate harder IWL scenarios, we gradually increase the number of training classes from 200 to 1600. We show quite bad and unstable ICL performance until 600 classes. ICL performance significantly improves for 1000 classes, but it is still unstable, with occasional ICL failure cases. However, we observe strong and stable ICL performance for a high number of classes as shown in Figure 17 where we report the strong ICL accuracies for the easier (2-way 4-shot) and harder (4-way 2-shot) setup. Impaired IWL accuracy for a higher number of classes shows the IWL task is now more difficult.

**Skewed distribution**    Keeping the total number of samples the same, we compare the ICL performance of a model trained with balanced and imbalanced (Zipfian) distributions. We observe improved ICL performance with skewed distribution, as illustrated in Figure 18. This experiment is a confirmation of prior work (Chan et al., 2022), which also shows improved ICL with the increased long-tail distribution.

We use three balanced datasets: 3600 samples across 200 classes, 7200 samples across 400 classes, and 10800 samples across 600 classes. These are compared to imbalanced datasets: 3598 samples across 463 classes, 7200 samples across 992 classes, and 10798 samples across 1551 classes, using a Zipfian distribution with a coefficient of 1.0. We observe improved ICL performance with skewed distribution, as illustrated in Figure 18. This experiment is a confirmation of prior work (Chan et al., 2022), which also shows improved ICL with the increased long-tail distribution.

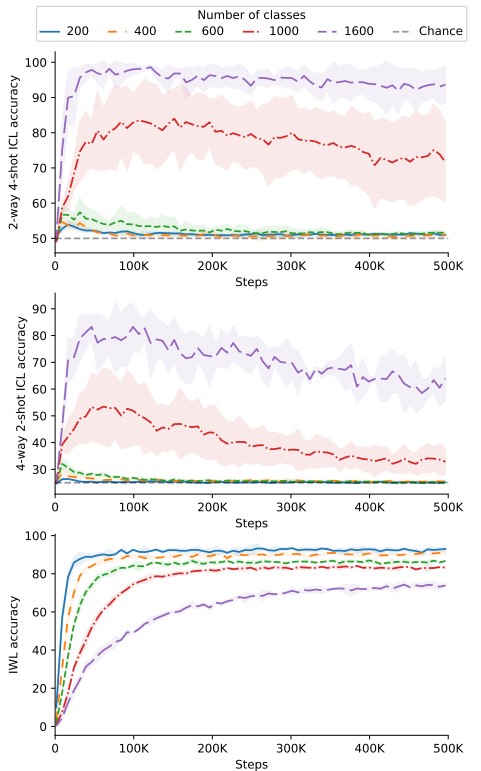

Figure 17: *Number of classes.* ICL (easier and harder setup) and IWL accuracy for different number of classes ranging from 200 to 1600. We can clearly notice the trend in stronger ICL ability with the harder IWL task, when we have more training classes.

Figure 18: *Skewed distribution.* ICL (easier and harder setup) and IWL accuracy for samples drawn from uniform and skewed distribution. By having the same number of training samples, we report better ICL performance with the skewed data distribution.

**Noisy labels**  We created two different baselines which represent the easier IWL setup with high IWL performance and non-existent or unstable ICL ability:

- using reduced number of classes (600)
- using 75% of in-context sequences and 25% of standard sequences

For each of the noise-label scenarios, we applied noise by randomly assigning a new label either to just a query label or to all labels in a sequence. There was no significant difference between the two methods, so we only present results for noise applied to the query label. We tested three different noise levels, ranging from 0 to 0.6 in increments of 0.2. Applying more noise to the standard sequences degrades the IWL performance while improving ICL performance for both cases of the experiment design (reduced number of class or reduced level percentage of the in-context sequences for training) as shown on Figures 7 and 19.

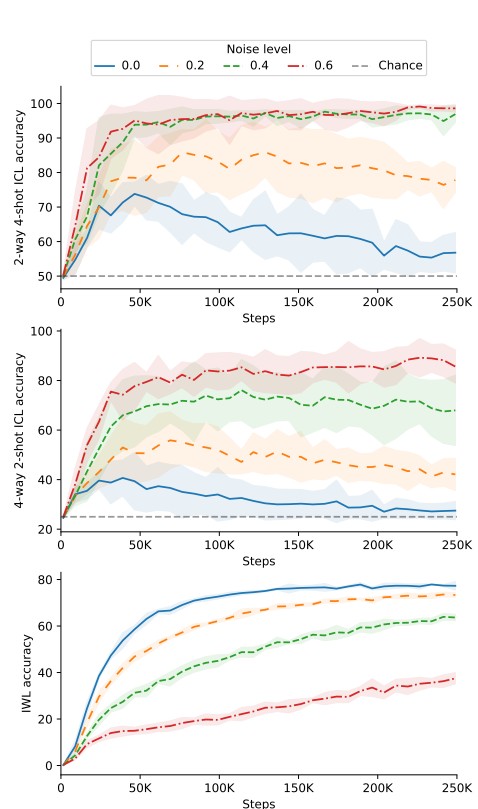

Figure 19: *Noisy labels.* Noisy scenario with reduced percentage of in-context sequences for training (75%). Adding noise improves ICL performance, but it is unstable with often ICL failure cases. Increasing noise in the training sequences also deteriorates the IWL performance since the IWL task is now harder.