# OpenReview forum: "What Matters for In-Context Learning: A Balancing Act of Look-up and In-Weight Learning"
_ICLR.cc/2025/Conference — Submitted to ICLR 2025_

### Official Review · Reviewer_w6Fy · 2024-11-02

**Soundness:** 3
**Presentation:** 3
**Contribution:** 3
**Rating:** 5
**Confidence:** 4

**Summary:**

LLMs have shown the capability to learn from context examples, while the mechanism behind ICL remains a black box. This paper explores the emergence of In-Context Learning capability of LLMs by introducing a controlled dataset and data sequences for the training process of a deep autoregressive model like GPT-2. The results show that conceptual repetitions in the data sequences are crucial for the ICL capability. Besides, the emergence of ICL also depends on balancing the in-weight learning objective with the in-context solving ability.

**Strengths:**

1.	The authors provide a new perspective for viewing the emergence of in-context learning capability, revealing the importance of conceptual repetition and IWL objectives.

2.	The proposed iCopy has proved effective in enhancing the in-context learning capability.

3.	The experiments are carried out across various vision datasets, which makes this paper more convincing.

**Weaknesses:**

1.	The presentation of this paper is not clear enough. For example, the sequence 3xQ-3xA-B-C is confusing initially. And the sequence (Q-A-B-C-D-E-F-G iCopy) also does not show which character symbol is repeated. Meanwhile, the use of “Q” and “A” could hint at the Q-A pairs, also increasing the difficulty of reading.

2.	Although the authors claim “conceptual repetitions” is crucial for the in-context learning capability, considering that “conceptual” also includes pure textual tokens, this paper does not conduct experiments on pure text tasks.

3.	Meanwhile, the real in-context learning tasks are much more complex compared to the settings in this paper. Actually, the selection of the examples will also influence the results due to the correlation between the examples and the current query.

**Questions:**

1.	Could the authors explain more about the formation of the sequences?

2.	Is there any experiments or evidence for reaching the same conclusions on pure text tasks?

3.	I am still concerned about the correlation between examples, which could influence the ICL prediction results. Do the authors agree with this point? If so, is there any method to evaluate the influence and try to relieve it?

---

> ### Author Response · Authors · 2024-11-21
>
> We thank the reviewer for their feedback and for raising important points. Below, we address your concerns in detail.
>
> **W1 and Q1:** We understand that the sequence notation may initially seem complex and appreciate the opportunity to explain it further. Each letter in the sequence notation refers to an image-label pair with the class label denoted by that letter. Query image token comes next after 8 image-label pairs as denoted in the sequence notation. The objective is to predict the class label of that query image. Q letter is specially used for image-label pairs in the sequence which have the same class label as the query image. Other letters (A, B, C…) denote other non-query image-label pairs. We acknowledge that having both Q and A in the notation might mislead the reader that we are working with QA tasks. We could change the notation to have classes A, B, C, D and only point out in the text that the query image and class are always coming from the class that has been repeated.
>
> We define training and evaluation classes. Training classes are used for generation of standard sequences and iCopy or non-iCopy sequences, while the evaluation classes are never seen during training and used only for ICL evaluation. The training classes are further split into train and valid split.
>
> The sequence without iCopy (3Q-3A-B-C) denotes a sequence where there are three image-label pairs that have the same class as the query class label but are different instances of that class. In the same sequence, there are also 3 image-label pairs with class label A that have different instances of that class. Whereas the sequence with iCopy (3Q-3A-B-C iCopy) denotes a sequence where there are again three pairs with the same class as query but are exact copies of the query image. The three pairs for class A are also copies of the same image. The locations of the image-label pairs are shuffled within the sequence to ensure the model does not learn the location bias. Sequences with iCopy and no-iCopy are only used for training to study the impact of repetitions in the sequence.
>
> Test sequences are the same for experiments with both sequences to ensure fair comparison. The sequences to test ICL have the query image always different from the images in the sequence and they use holdout classes which the model has not seen during training. ICL test sequences are constructed as either 2-way 4-shot (2 classes with 4 samples from each) or 4-way 2-shot with random baselines being 50% and 25% respectively.
>
> The sequences used to test IWL follow the same format as the standard sequence during training where all image-label pairs and the query in the sequence are unique. The used instances come from the same classes as the ones used for training, but on unseen instances from the validation split. In order to ensure fair comparison across different training settings, we predefined and precomputed the class split (training classes and holdout classes for ICL) and  validation sequences to ensure the difference in performance is because of the changes in training, and not evaluation bias.
>
> **W2 and Q2:** We believe experiments on textual data fall outside the scope of this study. Textual setups are inherently more complex due to grammar and syntax constraints, especially in the small model regime, making it challenging to design comparable experiments. Extending the study to large models, while intriguing, would introduce additional complexity, potentially obscuring our ability to test and confirm the hypotheses in a controlled manner.
>
> **W3 and Q3:** We performed the analysis when we were designing the study. We tested different seeds used to create the class splits and ICL sequences and we did not observe large differences in the results. In order to have a fair analysis, we have defined one class split and precomputed the sequences used for ICL and IWL validation so we can always compute the performance on the same sequences and see how exact modifications during training impact the performance.
>
> We hope this response clarifies your concerns and provides further insights into our experimental design. Thank you again for your thoughtful review and for engaging deeply with our work.

---

### Official Review · Reviewer_LnMe · 2024-11-03

**Soundness:** 2
**Presentation:** 3
**Contribution:** 1
**Rating:** 5
**Confidence:** 3

**Summary:**

In this paper the authors investigate the mechanisms behind In-Context Learning (ICL) in Large Language Models (LLMs). They argue that conceptual repetitions within training data sequences play a crucial role in enabling ICL. This view is contrasted to other explanations including the importance of data distribution and burstiness. The repetitions facilitate a "look-up" mechanism (essentially a copying mechanism performed by an inductive head) where the model learns to solve the task by recognizing the same pattern in the sequence and copying its associated label. The authors also highlight the significance of a complex in-weight learning objective in achieving stable and non-transient ICL performance. This complexity prevents the model from overfitting to simple patterns and encourages robust generalization.

**Strengths:**

1. The publication is sufficiently clearly written. While it does occasionally make qualitative statements without referring to any specific figure, section or result (for example, “We observe that it is not possible to obtain a clear ICL performance with the same architecture using only the high-burstiness strategy…”), it generally refers to empirical evaluations and has a sufficiently well thought out experimental section.
2. While being quite simple, the proposed techniques and ideas are sound. For example, it is entirely believable (even obvious and in the absence of careful experimental evidence) that copying a query example into the context would jump-start and incentivize the model to rely more on a simple “lookup” ICL mechanism.
3. The authors designed a reasonable set of experiments to study their proposed iCopy mechanism (as well as the effect of the label distribution, label noise and an instance discriminative task).

**Weaknesses:**

1. _Novelty_. In my opinion, this publication is quite incremental and it does not seem to make any significant conceptual leaps on top of the pre-existing work. The importance of the copying mechanism and understanding the role of inductive heads has been the subject of many prior publications. The central idea of the paper that label copying mechanism can be incentivized by directly copying the query sample is in my opinion too obvious. As one example, (Chan et al., 2022) has arguably made an intermediate step towards the iCopy setup by considering a dataset with a single Omniglot exemplar image. The present paper, among other things, does in some sense generalize this setup by allowing the context to contain the same exemplar image, while using many more exemplar images overall. However, I do not see this as a major leap.
2. It is hard to extend the proposed mechanisms beyond this simple few-shot learning setup. The proposed idea can be viewed as a form of data augmentation, where the sequence is enriched with the information necessary to promote a particular lookup mechanism. However, for real applications including most natural language tasks, this data augmentation (and in-context learning mechanism it empowers) are too naive. For example, rephrasing natural language sentences, changing their attributes, adding reasoning steps, etc. typically requires a separate teacher/oracle model and can be quite non-trivial and noisy [(Arthaud, 2021, Few-shot learning through contextual data augmentation), (Peng, 2023, Controllable Data Augmentation) and many more].
3. The simple copying mechanism studied here can actually be even damaging beyond the simple few-shot learning setup. In fact, there is ongoing work on “mitigating  the copying bias” (Ali, 2024, Mitigating Copy Bias in In-Context Learning through Neuron Pruning). Points 2 and 3 suggest to me that the iCopy mechanism may be a proper conceptual step for studying ICL in a simple few-shot learning setup, but may be inappropriate and difficult to re-interpret for most other complex in-context learning mechanisms.

**Questions:**

1. The core iCopy mechanism employed by the authors relies on a simple strategy: directly copying query examples and putting them into the context. This encourages the embedding model to produce reasonable sample embeddings and the Transformer model to perform ICL by attending to similar embeddings and replicating their associated labels. This may be suitable for some few-shot learning tasks with clearly defined sample embeddings, but this mechanism may be too simplistic for more complex forms of in-context learning including those necessary for most natural language tasks. There much more complex sequence processing and reasoning appears to be important. Relying on simple copying could even be damaging in these applications. In this light, what are potential tasks or applications where iCopy mechanism could actually be utilized to improve complex in-context learning capabilities?

---

> ### Author Response · Authors · 2024-11-21
>
> We thank the reviewer for their thoughtful feedback and pointed questions. Below, we address each of the raised concerns.
>
> **W1:** We appreciate the pointed out connection between our iCopy setup and the single exemplar setup by Chan et al. While we agree that there are some similarities, we believe our work diverges both in terms of objectives and outcomes. Chan et al. aim to show that larger within-class variation leads to improved ICL and show zero ICL performance with exact copies and shows a strong trade-off between ICL and IWL performance in other settings. In contrast, our setup shows strong ICL and IWL performance simultaneously advocating our argument - repetitions are crucial for ICL and they can be combined with IWL objective without harming IWL performance.
>
> **W2:** Our n-gram analysis (Figure 1) indicates that conceptual repetitions are naturally present in the natural text used for LLM training, supporting our hypothesis that these repetitions play an important role in achieving strong ICL performance. We agree with the reviewer that it might not be trivial to implement an analogous setup for natural language due to noisy data and multiple other dependencies. But we contend that exploring this idea in a controlled, simplified setup is important to isolate and understand the mechanism before extending this methodology to complex natural language tasks.
>
> **W3:** We believe copying bias is an understudied topic in LLMs with several factors at play. Therefore, it is hard to make conclusions. Our controlled study demonstrates that the iCopy data augmentation does not degrade performance but significantly enhances ICL performance across multiple settings.
>
> We hope these explanations provide clarity and context to our findings. Thank you again for your valuable feedback and engagement with our work.

---

> > ### Comment · Reviewer_LnMe · 2024-11-25
> >
> > I appreciate the authors' response to my questions. Their clarifications have provided a useful explanation of the perceived significance of their scientific contribution. While the generalizability of these findings to more complex scenarios remains to be seen (especially in the light of seemingly conflicting efforts to reduce copying bias in LLMs, something that the authors did not fully address in their reply), I acknowledge the value of this work in further illuminating the in-context learning mechanisms (and ways of improving this capability) within the specific few-shot learning context they've examined.

---

### Official Review · Reviewer_KpPA · 2024-11-04

**Soundness:** 2
**Presentation:** 3
**Contribution:** 2
**Rating:** 3
**Confidence:** 4

**Summary:**

- The authors analyze the relationship between conceptual repetition (e.g., burstiness, skewness) and ICL performance on vision-language few-shot learning tasks.
- The key take-aways are:
    - Simple repetition of the query (i.e. iCopy) in the given prompt/sequence of examples is sufficient for allowing ICL to emerge by developing in-context look up ability
    - Skewness, although helpful because they help create a more difficult IWL task, may not be necessary
    - A possible explanation of this phenomenon is through better construction of induction heads
- Given these key take-aways, the authors attempt to build on the data distributional properties for the emergence of ICL [1] and try to generalize these findings to the general LLM domain. However, this connection is weak and not supported by any experiments in the paper. it is not clear what iCopy may be equivalent to in the language domain and how these insights carry over to helping us understand the important properties of language pretraining.

[1] Chan, Stephanie, et al. "Data distributional properties drive emergent in-context learning in transformers." Advances in Neural Information Processing Systems 35 (2022): 18878-18891.

**Strengths:**

- The authors expand on the work of Chan et al. 2022 [1] to not only confirm their findings on the importance of the skewness and burstiness in the pretraining distribution for language, but also show that under the iCopy data construction regime, skewness may not be important and attempt to analyze why it is so (from the induction head perspective).

**Weaknesses:**

- The authors run experiments to investigate the relationship between conceptual repetition (e.g., burstiness, iCopy, etc) and ICL performance on vision-language few-shot learning tasks and try to generalize the observations to the LLM domain. However
- Some of the design choices for the experiments (like the baseline configuration, the objective function that only computes/backpropagates the loss on the query pair) are not justified and require clear explanation or empirical analysis. These questions are further expanded in the questions section.

**Questions:**

- How are the evaluation data samples constructed? (in both of the 2-shot and 4-shot settings)
- What’s the rationale behind why the authors choose the baseline model to be a mix of 10% and 90% of standard and in-context sequences? Is there perhaps an ablation on changing this ratio?
- When autoregressively trained on the Omniglot few-shot learning dataset, my understanding from the paper is that only the loss after 2L pairs of image-label examples is computed and used for training. Why do the authors choose this paradigm, instead of backpropagating the losses on all y_i along the sequence like in standard language modeling? How do the results change if we change the training objective to be autoregressive as well?
- In the experiments for Figure 6, with smaller # of classes, to achieve the same number of training steps, do you train over multiple epochs of the same dataset? Is it possible that these models are perhaps overfitting? Are there any training curves that we can analyze for this? Overall, it would be important to illustrate the validation performance on the IWL task and have that in the context of analyzing the ICL performance.
- This may be an experiment hard to run during the rebuttal, but how does the model size interplay in the ICL emergence with these different variables of burstiness, skewness, complexity of data (# of classes), etc, given that people have observed that ICL only emerges at some scale of model size?

---

> ### Author Response · Authors · 2024-11-21
>
> We thank the reviewer for their detailed feedback and thoughtful questions. Below, we address your concerns and provide further clarifications.
>
> **W1:** We appreciate your interest in extending this study to textual data. However, we believe such experiments are beyond the scope of this work. The setup with textual data presents additional complexities, especially in the small model regime, making it challenging to establish a comparable framework. While applying the study to larger models might seem feasible, it could significantly complicate the analysis and validation of our hypotheses, as it becomes harder to isolate and confirm the underlying mechanisms.
>
> **Q1:** For the ICL evaluation, we followed the methodology established in prior studies. Novel unseen classes were exclusively reserved for ICL evaluation, with no overlap with the classes seen during training. The base-novel class split was determined randomly to avoid selection bias. Few-shot ICL sequences were constructed either as 2-way 4-shot (2 classes, 4 samples per class) or 4-way 2-shot (4 classes, 2 samples per class), with random baselines at 25% and 50%, respectively. Importantly, no iCopy mechanism or copy-pasting was employed in the ICL evaluation sequences, as all images were unique instances. Including copy-pasting in ICL testing sequences would alter the nature of the task, turning it into a retrieval-based task rather than a classification problem.
>
> **Q2:** The mix of 90% in-context sequences and 10% standard sequences aligns with previous studies and serves as a well-established baseline for examining ICL performance. This proportion allows us to observe how changes in sequence structure affect performance while maintaining a clear and interpretable setup. For different mixes of in-context and standard sequences we observe the same ranking where sequences with iCopy performs better.
>
> **Q3:** The pretraining objective was designed to mimic autoregressive modeling, as is standard in language modeling. However, to adapt this approach for smaller models and explore the dependency on sequence format, necessary modifications were made, consistent with previous studies. When we applied a standard autoregressive loss to every intermediate token, the results were worse. This is because such an approach overemphasizes the intermediate item-label pairs, leading the model to learn misleading constraints. In language modeling, this approach works due to the inherent complexity of language, including strong grammatical and syntactical constraints. In our setup, however, this adaptation prevents the model from learning spurious patterns.
>
> **Q4:** We included IWL validation performance for the specified experiment in the Appendix (Figure 17).
>
> **Q5:** The Induction Head (IH) analysis was conducted on a smaller model to demonstrate that ICL can emerge under appropriate data distribution properties, even with limited capacity. We further tested a larger model (24 layers, 8 heads) and a smaller model (6 layers, 8 heads) to observe that smaller models achieve better IWL validation accuracy, whereas larger models excel in ICL performance. This observation aligns with our hypothesis that the difficulty of the IWL task matters as the model’s capacity plays an important role in this. We have not extended the analysis to even larger models to maintain the study’s focus on a smaller, more interpretable setup, which allows us to better isolate and analyze the effects of data properties.
>
> We hope these explanations clarify the methodology and contributions of our work. Thank you again for your thoughtful review and for engaging deeply with our research. We kindly request you to reconsider your evaluation in light of these clarifications.

---

> ### Comment · Reviewer_KpPA · 2024-11-25
> **Response**
>
> Thank you for the clarification on the experimental setup and the answers to the questions raised in the review.
>
> As for Q5, the insights sound interesting, but could the authors please upload a visualization/figure of the comparison between the three differently-sized models? It's hard to make any conclusions without seeing the results too.
>
> Also, throughout the paper, the authors still imply/indicate towards how these findings can generalize to large language models. However, not only is the pretraining objective (modified from the autoregressive objective where multiple query-label pairs are given first and not backpropagated on) different, but also it is still unclear how these can carry over in the natural setting. For example, the abstract's main claim is that "In this work, we systematically uncover properties present in LLMs that support the emergence of ICL." However, this statement rather seems to be over-claiming, as it is only a conjecture.
>
> Thus, I will currently stand with my original score.

---

### Official Review · Reviewer_cqA4 · 2024-11-07

**Soundness:** 2
**Presentation:** 1
**Contribution:** 3
**Rating:** 3
**Confidence:** 3

**Summary:**

This paper studies the origins of in-context learning (ICL) in a toy setup. Authors find that:

1. The presence of almost exact copies of a given sample in the model's context substantially promotes ICL.

2. ICL is diminished when the in-weight learning task is easy (as the model can solve the task without using context). On the other hand, harder tasks require both IWL and ICL, and hence the commonly observed ICL transiency (ICL disappearing as IWL performance increases) diminishes in harder tasks.

**Strengths:**

1. Timely topic and a relatively straightforward methodology for studying it.

2. I liked Figure 1 with plots of n-gram repetition in real corpora -- it's a good motivation for the toy setup in the paper. (I was keen to see a deeper investigation along those lines but only found toy experiments.)

3. The related work section is comprehensive, and I found it useful.

4. I liked the experiments in Section 5, though I'm not familiar enough with prior work to assess novelty.

**Weaknesses:**

The main weakness is that the core experiment in Section 4 is presented poorly, to the extent where I'm still not sure what precisely is going on even after going through the paper several times.

1. I don't get the difference between (3xQ-3xA-B-C) and (3xQ-3xA-B-C iCopy). Is the difference that in the iCopy setup, the three A and the three Q images in the context are copies of one another, whereas in the setting without iCopy they're different images with the same label? Also, does this only apply to the training data, or are test datasets also different between iCopy and no-iCopy?
    - a) More questions about iCopy description: What is conceptual repetition -- is it just different images of the same class, or is that exact repetition? What kinds of augmented versions of the images are used in iCopy?


    - b) It'd be helpful to clearly describe the difference between the baseline and iCopy settings -- perhaps by moving the baseline subsection into Section 4, or by adding a figure illustrating the difference.
    - c) Relatedly, I did not get an explanation in the Baseline subsection, from *>We think this happens ...* until the end of the paragraph.

2. I did not understand Figure 5: I'm not sure what to look for in these plots re induction heads -- how can the authors tell the induction head from the attention maps? I assume this has to do with the darker "columns" in subplot d? Also, I think it'd be helpful to add a description of induction heads to the related work section, as opposed to just mentioning them.


3. How is IWL performance measured? Is it zero shot?

**Questions:**

>In our setup, ICL transiency is eliminated by using repetitions in the data sequences and using a complex in-weight learning objective.

This sentence (with the context of the rest of the paper) makes me think that repetitions make ICL a relatively easier way to get low training loss compared to IWL. So you can get more ICL either by making ICL itself easier (repetition), or making IWL harder (your section 5) -- either way what matters is the balance of ease of ICL vs ease of IWL, and the usefulness of each for reducing the training loss.

Does the above seem right to the authors? If so, highlighting this point about the balance between ICL and IWL difficulty could be helpful, as it seems to unify many of your findings.

---

> ### Author Response · Authors · 2024-11-21
>
> We sincerely thank the reviewer for their insightful feedback and valuable suggestions. Below, we address the key concerns and provide additional clarifications.
>
> **W1 and W3.** We understand that the sequence notation may initially seem complex and appreciate the opportunity to explain it further. Each letter in the sequence notation refers to an image-label pair with the class label denoted by that letter. Query image token comes next after 8 image-label pairs as denoted in the sequence notation. The objective is to predict the class label of that query image. Q letter is specially used for image-label pairs in the sequence which have the same class label as the query image. Other letters (A, B, C…) denote image-label pairs of other non-query classes.
>
> The sequence without iCopy (3Q-3A-B-C) denotes a sequence where there are three image-label pairs that have the same class as the query class label but are different instances of that class. In the same sequence, there are also 3 image-label pairs with class label A that have different instances of that class. Whereas the sequence with iCopy (3Q-3A-B-C iCopy) denotes a sequence where there are again three pairs with the same class as query but are exact copies of the query image. The three pairs for class A are also copies of the same image. Sequences with iCopy and no-iCopy are only used for training to study the impact of repetitions in the sequence.
> Test sequences are the same for experiments with both sequences. The sequences to test ICL have the query image always different from the images in the sequence. ICL test sequences are constructed as either 2-way 4-shot (2 classes with 4 samples from each) or 4-way 2-shot with instances from unseen classes to evaluate ICL performance.
>
> The sequences used to test IWL follow the same format as the standard sequence during training where all image-label pairs and the query in the sequence are unique. The used instances come from the same classes as the ones used for training, but on unseen instances from the validation split.
>
> Conceptual repetitions in the Omniglot setup refers to the iCopy sequences where there are exact instance copies of query images present in the sequence. There are no augmentations used in this setup. For other visual datasets, we performed mild augmentations by rotation and cropping as it has shown to improve the performance slightly.
>
> **W2.** Induction heads [1] represent foundational circuits essential for the emergence of in-context learning [2].  We examined how similar circuits operate in our setup by looking into the attention maps of the given sequence in different layers for two models - one exhibiting ICL and one having no ICL abilities. Formation of the induction head on Figure 5 can be observed only on a model trained with iCopy sequences. The induction head is a two layer circuit where in the first layer we can notice a pattern on the diagonal moved by one (dark colors on the figure) which represents label tokens attending to the previous image tokens. Next, in the second layer, we can observe that the query image token attends to the positions of the image-label pairs which have the same label as the query. We provide more explanation and additional graphs on this analysis in the appendix where we also report different progress measures (averaged values of attention scores) to show the behavior of induction heads.
>
> **Q1.** The iCopy ensures a stronger and reliable look-up mechanism which eventually leads to better ICL performance. The ICL task with this modification is not easier, but harder because the ICL evaluation sequences do not have any copy-pasted images in the context since the sequence consists of 9 unique images. Thus, the iCopy sequence does not make the ICL task easier, but it ensures better pattern matching which is needed for the novel classes and unique images during inference.
>
> We hope this explanation clarifies the concerns raised and provides additional context for our experimental design. Thank you once again for engaging deeply with our work. We kindly ask you to reconsider your initial evaluation in light of these points.
>
> References:
> - [1] https://transformer-circuits.pub/2022/in-context-learning-and-induction-heads/index.html
> - [2] https://openreview.net/forum?id=aN4Jf6Cx69

---

> > ### Comment · Reviewer_cqA4 · 2024-11-24
> >
> > Thank you for the detailed response! Re W1 and W3, your setup is now clear to me. Nevertheless, I will maintain my score as I believe the paper would substantially benefit from edits making it more accessible in the first place.
> >
> > Re Q1: you are correct that at test time, the ICL task is harder for the model trained with iCopy (since the test task a bit OOD for it). What I meant is that iCopy makes ICL an easier strategy to achieve low loss *during training*.

---

### Meta-Review · Area_Chair_qfQR · 2024-12-17

**Metareview:**

This paper has received ratings of  5, 5, 3, 3 and was unanimously recommended for rejection by all reviewers.

This papre explores the mechanisms underlying In-Context Learning (ICL) in a controlled, toy setup using an autoregressive GPT-2 model for image classification tasks. The authors identify that repetitions of image-label pairs (referred to as iCopy) in training data sequences play a significant role in enabling ICL, outperforming prior focuses on burstiness or skewed distributions. The authors further show that ICL emergence corresponds to the formation of induction heads, where repetitions facilitate a reliable look-up mechanism. Experiments were conducted on controlled datasets (Omniglot, CIFAR-100, Caltech-101, DTD), producing results that support the proposed hypotheses.

Strengths
- Timely topic: The paper addresses an important, open question in understanding ICL mechanisms, aligning with recent advancements in generative models.
- Well-motivated hypotheses: The study builds on prior works but differentiates itself by disambiguating the role of repetitions versus burstiness and other data properties.

Area for improvements:
- The clarity in its experimental setup and sequence notations can be further improved. For instance, the differences between "iCopy" and baseline sequences were confusing to reviewers, hindering the readability of core experiments (Section 4). Improvements in organization and illustrative figures are necessary.
- Limited Generalization to LLMs: While the paper speculates on connections to Large Language Models, it lacks empirical validation in the language domain. This limits the broader applicability of its claims.
- The contributions seem to be incremental. Several reviewers noted that the idea of repetitions enhancing ICL (through look-up mechanisms) feels intuitive and builds incrementally on prior work (e.g., Chan et al., 2022). The novelty of the findings may not justify acceptance at a top-tier venue.
- Insufficient justification for design choices: Questions about training ratios (e.g., 90% in-context sequences) and modified objectives remain inadequately addressed. Clearer explanations or ablations are needed.

The reviewers consistently appreciated the focus on understanding ICL and the controlled methodology. However, there were shared concerns about:
- The clarity of presentation, particularly in describing sequences and experimental design.
- The generalizability of findings to natural language tasks, as claims regarding LLMs remain speculative.
- The incremental nature of the contributions relative to existing works on burstiness and data properties for ICL.

While the paper tackles a timely topic and provides some interesting insights, the unclear presentation, incremental contributions, and lack of empirical validation on natural language tasks undermine its overall impact. The paper requires significant restructuring and expansion to address these issues before being considered for acceptance at ICLR.

**Additional Comments On Reviewer Discussion:**

Despite the authors' detailed rebuttals addressing concerns (e.g., clarifying iCopy, sequence construction), reviewers maintained their original scores, indicating that the paper would benefit significantly from further revisions to improve clarity and strengthen generalizability.

---

### Decision · Program_Chairs · 2025-01-22

Reject